# Clinical translational research of liquid biopsy applications in prostate cancer and other urological cancers

Jingyi Huang[1] (iD), Da Huang[1], Xiaohao Ruan[1], Yongle Zhan[2], Stacia T.-T. Chun[2], Ada T.-L. Ng[3] and Rong Na[2] (iD)

[1]Department of Urology, Ruijin Hospital, Shanghai Jiao Tong University School of Medicine, Shanghai , China; [2]Department of Surgery, School of Clinical Medicine, The University of Hong Kong, Hong Kong, Hong Kong and [3]Division of Urology, Department of Surgery, Queen Mary Hospital, The University of Hong Kong, Hong Kong, Hong Kong

## Review

**Keywords:**
liquid biopsy; prostate cancer; circulating tumor cell; cell-free DNA; extracellular vesicle

**Corresponding author:**
Rong Na;
Email: narong.hs@gmail.com

## Abstract

The aim of liquid biopsies is to obtain tumor information via the molecular interrogation of liquid samples, including blood and urine. As a minimally invasive procedure, liquid biopsies have attracted attention. A series of studies have reported associations of biomarkers such as circulating tumor DNA, cell-free DNA and extracellular vesicles with urological cancers, especially prostate cancer (PCa), and demonstrated the promising potential of liquid biopsies. In this review, we summarize recent clinical translational studies of liquid biopsies in PCa and other urological cancers, including bladder cancer and renal cell carcinoma. The number of translational studies was limited, and most of the studies focused on PCa. Biomarkers isolated from blood by different detection methods could be applied in clinical practice to predict prognosis and treatment response in advanced PCa. The other applications in urological cancers identified in previous studies remain to be explored further. Current studies are limited due to the lack of ideal standard detection methods for biomarkers. In the future, with advances in methodology, more translational studies will be conducted to identify potential applications of liquid biopsies in urological cancers.

## Impact statement

Liquid biopsies have promising potential to ameliorate cancer diagnosis and treatment, whose aim is to obtain tumor information via the molecular interrogation of liquid samples, including blood and urine. As minimally invasive procedures, liquid biopsies have attracted attention. This review summarizes current clinical translational studies of liquid biopsy in PCa and other urological cancers, involving circulating tumor DNA, cell-free DNA and extracellular vesicles. Biomarkers isolated from blood by different detection methods could be applied in clinical practice to predict prognosis and treatment response in advanced PCa. The other applications in urological cancers identified in previous studies remain to be explored further. Current studies are limited due to the lack of ideal standard detection methods for biomarkers, which deserves further exploration.

## Introduction

The concept of the liquid biopsy was first raised about 10 years ago when circulating tumor cell (CTC) was introduced (Pantel and Alix-Panabières, 2010). This technique aims to obtain tumor-derived information via the molecular interrogation of liquid samples, including blood, urine and so on (Nikanjam et al., 2022). With the rapid development of gene detection and cell separation technologies, the potential of liquid biopsies is constantly being explored. The factors detected in liquid biopsies include circulating tumor DNA (ctDNA), cell-free DNA (cfDNA), tumor-educated platelets and extracellular vesicles (EVs) (Nikanjam et al., 2022).

Compared with traditional tissue biopsies, liquid biopsies are minimally invasive procedures and available at all status of disease. From a pathological point of view, liquid biomarkers provide an evaluation of the entire tumor, while tissue biopsy only involves part of the tumor, which may lead to severe bias due to tumor heterogeneity (Cimadamore et al., 2019). Liquid biopsies are expected to replace some invasive examinations and to be applied at various stages during cancer diagnosis, follow-up and treatment.

For urologic cancers, liquid biopsies have also emerged as a promising strategy, especially for prostate cancer (PCa), bladder cancer (BCa) and renal cell carcinoma (RCC). Several blood and

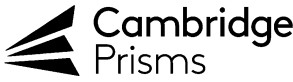

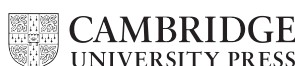

urine biomarkers have been demonstrated to be related to risk or prognosis in PCa (Crocetto et al., 2022b). For example, small RNAs carried in urinary EVs might be useful for PCa diagnosis (Rönnau et al., 2014). Increased CTCs in blood after chemotherapy might be related to poor response to treatment (Zapatero et al., 2020). Urinary biomarkers were more comprehensively studied in BCa because of their direct contact with tumor and their isolation from homeostasis (Crocetto et al., 2022a). A series of markers related to disease outcome have been revealed based on analyses of genetic abnormalities and epigenetic alterations, such as changes in urine DNA methylation, proteomic and mutation profiles (Matuszczak et al., 2022). For RCC, both blood and urine biomarkers have been found to be valuable for predicting progression risk and indicating the efficiency of treatment (Li et al., 2023).

An increasing number of biomarkers were proposed and discovered as the liquid biopsy concept is getting more and more important in recent years. However, a few of them were actually applied in clinical management due to the lack of translational research evidence to indicate how to use liquid biopsies to ameliorate clinical strategy. Translational research functions as a bridge between basic research and clinical practice (Dragani et al., 2016). To promote the application of liquid biopsies, several translational studies have been carried out in different diseases, such as hepatobiliary cancer, lung cancer and urological cancers (von Felden et al., 2020; Malapelle et al., 2021).

In this review, we summarize the present studies regarding the clinical translational researches for the application of liquid biopsies on urological cancers, including PCa, BCa and RCC. We aimed to cover the studies with clinical application potentials. We also delineated challenges and perspectives of future development of translational research for liquid biopsies.

## Evidence synthesis

We searched published research articles using the keywords "liquid biopsy", "translational research" and urological cancers (including "PCa," "BCa," and "RCC") in the PubMed and Embase databases. To refine the search and make it comprehensive, "liquid biopsy" was replaced by "CTC," "ctDNA," "cfDNA," "EV" and "exosome." A total of 57 results were found. After review of the abstracts and full articles, 10 original research articles that were confirmed to be clinical translational research studies were ultimately included in this review (Table 1). In addition, 13 original research articles reported the translational potential of liquid biopsies.

**Table 1.** Summary of clinical translational studies of liquid biopsy for PCa

| Study | Biomarkers | Liquid | Detection | Patients | Conclusions |
|---|---|---|---|---|---|
| Mandel et al., 2021 | CTC | Blood | CellSearch | 33 mCRPC | CTC enumeration contributes to prognostic information |
| Oeyen et al., 2021 | CTC and EV | Blood | CellSearch + ACCEPT | 170 mCRPC | Automated CTC counts are a reliable substitute for reviewer-based enumeration, as they are equally informative for prognosis assessment in patients with mCRPC. EVs are predictive markers for prognosis of mCRPC |
| Lindsay et al., 2016 | CTC | Blood | CellSearch | 156 mCRPC | Poorer survival outcomes were observed in vimentin- and Ki67-positive CTC patients |
| Di Lorenzo et al., 2021 | CTC | Blood | Flow cytometry | 53 mCRPC, 5 control | Flow cytometry could be used to enumerate CTCs, but also to assess molecular biomarkers on their surface.AR-V7 + CTC count was also associated with worse outcome |
| Carles et al., 2018 | CTC | Blood | Spanish Society of Medical Oncology | 45 mCRPC | CTCs hold significant promise as a prognostic factor for survival and completing treatment prior to the initiation of bone-targeted radium-223 therapy |
| Choudhury et al., 2018 | cfDNA | Blood | ichorCNA | 140 mCRPC | Tumor fraction correlates with clinical features associated with overall survival (OS) in CRPC, and its decline is a promising biomarker for initial therapeutic response |
| Colosini et al., 2022 | cfDNA | Blood | Deep target sequencing approach | 28 oligometastatic PCa | Recognized the heterogeneity of oligometastatic prostate disease, and a better molecular characterization of oligometastatic disease. CfDNA predicted an eventual polymetastatic progression |
| Del Re et al., 2017 | EV | Blood | Highly sensitive digital droplet polymerase chain reaction (ddPCR) | 36 CRPC | Plasma-derived exosomal RNA is a reliable source of AR-V7 that can be detected sensitively by ddPCR assay. Resistance to hormonal therapy may be predicted by AR-V7, making it a clinically relevant biomarker |
| Kim et al., 2022 | EV | Blood, urine | Flow cytometry | 35 BPH, 85 local PCa, 20 mPCa | Proof-of-feasibility study can serve as a scientific and technical framework for other groups motivated in using flow cytometry for EV research |
| Signore et al., 2021 | EV | Blood | Antibody-based proteomic technology + ultracentrifugation | 19 primary PCa, 6 CRPC | Serum-derived EV cargo may be exploited to improve the current diagnostic procedures while providing potential prognostic and predictive information |

ACCEPT, automated CTC Classification Enumeration and PhenoTyping; cfDNA, cell-free DNA; CTC, circulating tumor cell; ddPCR, droplet digital PCR; EV, extracellular vesicle; mCRPC, metastatic castrate-resistant prostate cancer.

## Enumeration and assessment of CTCs

CTCs are cancer cells separated from the primary tumor or metastatic foci. CTCs are rare; at most, one tumor cell can be found in a hundred million cells circulating in the blood (Nelson, 2010). Given the broad range of technologies used to capture CTCs, CTC detection has become more efficient; thus, various clinical translational studies of CTCs have been conducted. Presently, CellSearch (Menarini Silicon Biosystems) is the only clinical CTC enumeration method that has received Food and Drug Administration (FDA) approval (Riethdorf et al., 2018).

The number of CTCs in blood determined by flow cytometry might predict the prognosis of metastatic castration-resistant prostate cancer (mCRPC). Higher CTC counts were associated with worse radiographic progression-free survival and OS according to the results of translational research (Di Lorenzo et al., 2021). Similarly, for patients with mCRPC starting abiraterone acetate or enzalutamide therapy, higher CTCs at baseline and an increase in CTCs at follow-up were independent predictors of worse prognosis (De Laere et al., 2018). A few years later, the same group reanalyzed the previous images using an image analysis tool called the automated CTC Classification Enumeration and PhenoTyping (ACCEPT) tool (Oeyen et al., 2021). The results showed that beyond enumeration, high CTC phenotypic heterogeneity was also related to worse survival outcome in mCRPC. With regard to the CTC phenotype, in 2016, Lindsay et al. established a routine for the clinical testing of Ki67 and vimentin expression in CTCs and demonstrated that the presence of Ki67 and vimentin expression in CTCs was related with poor outcome in mCRPC (Lindsay et al., 2016). Peripheral blood (7.5 mL) from patients with mCRPC was collected and tested for Ki67 and vimentin. The results showed a significant reduction in OS in patients with Ki67- or vimentin-positive CTCs. It is the presence of expression but not the exact number or proportion of Ki67- or vimentin-positive CTCs which was associated with OS. In addition to mCRPC, Mandel et al. focused on oligometastatic hormone-sensitive prostate cancer (HSPC), an earlier stage of PCA. For patients with oligometastatic HSPC who underwent cytoreductive radical prostatectomy, higher CTC numbers both before and 6 months after surgery indicated shorter PFS to CRPC. Interestingly, with regard to the prognostic value, CTC number was a better biomarker than lactate dehydrogenase, prostate-specific antigen (PSA) and bone-specific alkaline phosphatase (Mandel et al., 2021).

Assessment of CTCs has also been used to predict treatment response. Abiraterone is a novel first-line hormonal therapy, and enzalutamide, an oral second-generation anti-androgen receptor (AR), was shown to be effective for patients with mCRPC before or after chemotherapy. A prospective, multicenter, translational study was conducted and showed that a higher androgen receptor splice variant 7 (AR-V7) positive CTC count in blood predicted a poorer response to enzalutamide (Di Lorenzo et al., 2021). Furthermore, Antonarakis et al. (2017) categorized samples from 202 patients with mCRPC receiving abiraterone or enzalutamide into three groups: CTC−, CTC+/AR-V7- and CTC+/AR-V7+. It was demonstrated that the treatment response and survival were the best for CTC− patients and the worst for CTC+/AR-V7+ patients.

In addition, the Spanish Oncology Genito-Urinary Group performed a prospective translational study of a cohort of 45 patients with mCRPC who were treated with radium-223 (Carles et al., 2018). The results indicated that patients with CTC counts ≤5/7.5 mL before radium-223 treatment responded better and were more likely to continue therapy. Additionally, the survival outcomes were better in patients with few CTC counts.

Few translational studies of CTCs in RCC and BCa exist. It was supposed that the lack of translational evidence about CTC in RCC was probably owing to limitations related to the CTC isolation method. The CellSearch system mentioned above is based on the detection of epithelial cell adhesion molecule (EpCAM) and cytokeratin. However, RCC tumor cells have low EpCAM and low cytokeratin expression. Therefore, the standard methods of CTC detection might not be optimal for RCC (Li et al., 2023). Cappelletti et al. (2020) reported an efficient, marker-independent approach that assesses the expression of *EpCAM*, *MUC1* and *ERBB2* to detect CTCs in RCC. Although the predictive value of CTCs in RCC was not clear in this study due to the limited sample size, it was confirmed that the existence of CTCs with epithelial markers was correlated with shorter PFS.

Some clinical translational studies have explored CTCs, the first marker derived from liquid biopsies, to identify their predictive effect in urologic cancers. Enumerating CTCs in blood and assessing the expression of certain biomarkers can aid prognosis prediction and the development of personalized therapy in mCRPC and mHSPC. Few studies have illustrated the predictive effect of CTCs in RCC and BCa. Additionally, all the existing studies have used blood.

## Cell-free DNA and circulation tumor DNA

Circulating cfDNA molecules are extracellular, short fragments of nucleic acids circulating in body fluids and are approximately 134–144 bp (Lu et al., 2019). ctDNA is a type of cfDNA derived from tumor cells. The quantity of ctDNA depends on the overall tumor volume. With the development of ddPCR and next-generation sequencing, cfDNA can be very sensitively quantified such that even rare mutations and recently methylated sequences can be identified (Poulet et al., 2019). cfDNA detection has potential applications in disease screening, prognostic prediction and evaluation of treatment response.

Similar to CTCs, ctDNAs were explored more in PCa among urological cancers by translational studies. The concentration and fragment size of seminal cfDNA were significantly higher in PCa patients than benign prostate hyperplasia (BPH) patients or healthy controls, which indicated that seminal cfDNA might function as a screening biomarker (Poulet et al., 2019). Furthermore, ctDNA could also be used to evaluate the prognosis of castration-resistant prostate cancer (CRPC). An analysis of 663 plasma samples from 140 CRPC patients showed that ctDNA was related to worse survival outcome (Choudhury et al., 2018). Interestingly, the proportion of ctDNA was positively correlated with PSA and ALP and negatively correlated with hemoglobin. It was reported that methylation of *SRD5A2* and *CYP11A1* DNA in blood cfDNA was associated with a higher risk of biochemical recurrence in patients with HSPC after radical prostatectomy (Horning et al., 2015).

Regarding value for predicting therapy response, a prospective translational study was conducted in a cohort of 28 patients with oligometastatic HSPC undergoing stereotactic body radiotherapy (SBRT) (Colosini et al., 2022). Deep targeted cfDNA sequencing analysis revealed the molecular heterogeneity of metastatic PCa and helped to identify the exact characteristics to develop a specific therapeutic strategy. For instance, *BRCA1* mutation in ctDNA was associated with SBRT failure.

The application of ctDNAs presented translational potential in clinical management of BCa and RCC. Shallow-depth bisulfite sequencing of urinary cfDNA by methylation deconvolution and

analysis of global hypomethylation and copy number aberrations (CNAs) has been proposed as a method to detect BCa, with a sensitivity of 93.5% and a specificity of 95.8% (Cheng et al., 2019). Furthermore, the levels of methylation and CNAs might reflect disease stage. A multiomic urinary cfDNA analysis strategy proposed by Chauhan et al. was also proven to be capable of detecting molecular residual disease of BCa (Chauhan et al., 2023). Additionally, for patients with metastatic clear cell RCC, lower cfDNA levels in the bloodstream were associated with a better response to treatment and longer PFS, but further study is required for confirmation (Del Re et al., 2022).

In summary, cfDNA has broad diagnostic and prognostic utility mainly for PCa and BCa. The sources of liquid samples in studies have varied greatly, including blood, seminal fluid and urine. Additionally, there are many aspects of cfDNA that are worth exploring, such as cfDNA quantity, mutations, and methylation patterns.

### Extracellular vesicles and exosomes

EVs are lipid bilayer membrane-enclosed nanometer to micrometer vesicles secreted continuously by almost all mammalian cells into the extracellular space (Théry et al., 2018). They were first regarded as a means for cells to release waste. However, it was proven that EVs play an important role in intracellular communication. EVs are very heterogeneous in terms of size and characteristics, with various sizes, methods of biogenesis, origins, biological characteristics and release mechanisms. Exosomes are the most commonly studied subgroup of EVs of endocytic origin, and they range in size from 30 to 100 nm (Becker et al., 2016), while oncosomes, as larger vesicles, can be up to 10 μm in diameter. EVs exist in different biological fluids, such as serum, plasma, urine and saliva, with various cargos, such as DNA, RNA, proteins and lipids. MicroRNAs (miRNAs) are the most frequent cargo. Tumor cells were found to release more EVs than other cells (Li et al., 2018). Recently, EVs, especially exosomes, have been proven to contain tumor information and function in the progression and metastasis of cancer (Casanova-Salas et al., 2021). The potential of EVs as diagnostic and prognostic markers has attracted much attention.

Centrifugation-based approaches are applied to isolate EVs, and ultracentrifugation is considered the standard method (Théry et al., 2018). However, the efficiency is disagreeable due to its low throughput and poor specificity (Gerdtsson et al., 2021). To improve the efficiency and specificity, several methods have been explored for the detection and characterization of EVs.

Three translational studies on the use of exosomes in PCa were identified with the present search strategy. Each of the studies developed EV measurement methods. Kim et al. (2022) first proposed a systematic method using standardized and calibrated flow cytometry for the quantification of PCa-derived EVs from blood and urine with detailed methodologies, instrument characteristics and acquisition settings. Their study validated the effectiveness of this method and demonstrated that PSMA+ EVs and STEAP+ EVs might be surrogate markers for metastatic PCa. Reverse-phase protein microarrays, as an antibody-based proteomic technology, have also been applied to analyze EV cargo and were used to identify a series of potential biomarkers for EV-based PCa diagnosis, such as PD-1 and survivin (Signore et al., 2021). The last translational study was mentioned in the CTC section. When Oeyen et al. (2021) reanalyzed CellSearch images of patients with mCRPC with ACCEPT software, EVs were efficiently detected as well. It was demonstrated that a higher baseline EV level was

associated with shorter PFS of patient with mCRPC, which is in line with the prognostic value of CTCs.

In addition to the translational studies mentioned above, evidence from a number of nonexplicit translational studies was reviewed, also supporting the translational value of EVs and exosomes. As early as 2012, the prognostic value of exosomes in PCa was explored. After isolation by centrifugation, the number of plasma exosomes in PCa patients was significantly higher than that in BPH patients or healthy controls. Survivin is an inhibitor of apoptosis family proteins that is associated with PCa development, and exosomal survivin was also found to be higher in PCa plasma. No difference was observed between patients with PCa with high and low Gleason scores (Khan et al., 2012) However, the number of plasma prostate microparticles, a kind of EV, could identify PCa patients with Gleason score $\geq 4 + 4$ by nanoscale flow cytometry (Biggs et al., 2016). In addition to their screening value, EVs might also serve as prognostic biomarkers. Two blood-based exosomal miRNAs (miR-1290 and miR-375) were identified as biomarkers that could predict the prognosis of mCRPC (Huang et al., 2015). In addition, significant alteration of exosomal miRNA and protein cargos was observed as neuroendocrine differentiation occurred in PCa, and neuroendocrine prostate cancer has poor survival outcomes with limited treatment options (Bhagirath et al., 2021). With regard to value for predicting treatment response, AR-V7 carried in blood exosomes could serve as a reliable biomarker of resistance to hormonal therapy (Del Re et al., 2017).

Few definite translational studies have focused on EVs in BCa or RCC, though EVs have translational value for these two cancers (Zeuschner et al., 2020). For example, exosomal miR-1233 and miR-210 in blood were significantly increased in RCC patients compared with healthy controls (Zhang et al., 2018). Regarding the prognosis of RCC, higher miR-224 levels in serum EVs were related to poorer survival outcomes (Du et al., 2017). For BCa, different studies showed that a series of long noncoding RNAs were increased in urinary EVs of BCa patients compared with healthy controls (Zhan et al., 2018; Xue et al., 2021), while several cargo molecules in blood EVs, such as LNMAT2, could predict the prognosis of advanced BCa (Becker et al., 2016).

In summary, EVs and exosomes have been well studied and shown to have strong translational potential for screening and predicting outcome. However, to date, a standard methodology with high efficiency and sensitivity to detect EVs is still lacking.

### Perspective and future directions

In the past decade, a variety of studies have revealed the association between the development of urological cancer and factors detected in liquid biopsies, including CTCs, cfDNA and EVs. Undoubtedly, liquid biopsies can be used to detect promising biomarkers that can be applied for disease diagnosis and prognosis and treatment response prediction in PCa and other urological cancers. Currently, traditional tissue biopsy is still the gold standard for disease diagnosis and serves to guide the therapy strategy. Compared with tissue biopsies, liquid biopsies have their own advantages. First, due to tumor heterogeneity, tissue biopsies may not reflect the comprehensive situation of the disease, while liquid biopsies can help to provide a general assessment. Second, with advances in clinical practice, it is important to avoid unnecessary invasive procedures. The application of liquid biopsies may greatly reduce the number of invasive biopsies. For example, because of the limited specificity of PSA as a screening biomarker for PCa, some patients undergo unnecessary prostate biopsy (Duffy, 2020). The clinical

application of liquid biopsies when appropriate would efficiently ameliorate this situation. Second, the liquid biopsy is a minimally invasive procedure with a relatively rapid and simple process, making it a better option for patients. Liquid biopsies are also generally less of an economic and socioeconomic burden than tissue biopsies. Third, liquid biopsies can be performed at all stages and thus can be used to stratify patients after radical surgery. Dynamic assessment of liquid biomarkers can be used to guide long-term clinical decisions.

Clinical translational studies function as a bridge between basic studies and clinical use, providing supporting evidence and directing actual application. We reviewed clinical translational studies of liquid biopsies in PCa, BCa and RCC. As the most frequent urological cancer, PCa has received the most attention. In PCa, CTCs and cfDNA from blood have been shown to be able to predict survival outcome and therapy response in CRPC and metastatic HSPC. Interestingly, most studies on CTCs have used the CellSearch platform, which was approved by the FDA as a method to detect CTCs, while two translational studies on cfDNA employed different methods for detection. Studies of EVs have used various detection methods.

To date, the number of specific clinical translational studies is limited and further exploration is warranted. The lack of an ideal standard method for the isolation of targeted biomarkers is one of the main reasons for the limited performance in relevant translational studies. As mentioned above, the CellSearch system is an FDA-approved method for detecting CTCs. However, it is expensive and not suitable for certain cancers, such as RCC (Li et al., 2023). With regard to cfDNA and EVs, there is also no standardized method. Without a standardized detection protocol, it is difficult to assess the true utility of liquid biopsies and to apply relevant strategies in the clinic. Therefore, it is essential to explore standardized methods and comprehensive protocols for the identification of different biomarkers. In addition, recent translational research has not explored the utility of liquid biopsy analysis for the selection of therapy. For example, the expression of AR-V7 in blood CTCs or EVs was found to be associated with resistance to second-generation anti-androgen receptor and hormone therapy (Del Re et al., 2017; Oeyen et al., 2021). Therefore, it remains to be determined whether the expression of AR-V7 can guide the selection of second-generation anti-androgen receptor therapy or chemotherapy. Finally, there is still plenty of potential for translational research on liquid biopsies. Recent translational studies have focused on biomarkers in blood, while other fluid biomarkers may also be valuable. Seminal cfDNA levels are much higher (almost 100 times higher) than blood cfDNA levels in PCa patients (Ponti et al., 2021). Urinary biomarkers might have potential in BCa because they have direct contact with tumors.

## Conclusion

The strong potential of liquid biopsy factors, including CTCs, cfDNA and EVs, in urological cancer, especially PCa, has been proven by a series of clinical translational studies. As assessed by different detection methods, biomarkers isolated from blood can be applied in clinical practice to predict prognosis and treatment response in advanced PCa. Other applications of liquid biopsies in urological cancers remain to be further explored based on the results of previous studies.

**Open peer review.** To view the open peer review materials for this article, please visit http://doi.org/10.1017/pcm.2023.19.

**Author contribution.** Conception and design: R.N., D.H., J.H., S.T.-T.C., A.T.-L.N.; Literature review: J.H., D.H., X.R. Y.Z.; Manuscript writing: R.N., J.H., D.H., X.R.

**Competing interest.** The authors declare none.

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
