## [Reviewer Report]

The manuscript entitled "Clinical translational research of liquid biopsy applications

in prostate cancer and other urological cancers" highlighted that Current studies are limited due to the lack of ideal standard detection methods for biomarkers. In the future, with advances in methodology, more translational studies will be conducted to identify potential applications of liquid biopsy in urological cancers.

Minor comments:

- The AUthors should provide the expand forms for all acronyms, i

---

## [Reviewer Report]

Rong et al. wrote this review to discuss and summarize available clinical translational research of liquid biopsies in urological cancers, specifically prostate cancer (PCa), bladder cancer (BCa), and renal cell carcinoma (RCC). The review explains liquid biopsies, the different materials that can be used, and the different types of biomarkers that can be found in liquid biopsies. The main body of the review is organized by three big biomarkers found in liquid biopsies. The authors explain their methods for finding the clinical translational research, summarize/connect the findings of those papers, and provide what they believe to be the next steps the scientific community should take to further bridge the basic studies to clinical use of liquid biopsies.

In my opinion, this review is well-organized and well-written. It provides a thorough explanation of the topic at hand. The review’s conclusion and future directions are relevant and necessary to further the research in liquid biopsies. I would accept this review after some suggested minor revisions. The revisions mainly pertain to the structure of paragraphs and the grammar of the paper.

Some revisions are:

• “Liquid Biopsy” is sometimes singular and sometimes plural. In most cases/sentences, I believe “liquid biopsy” should be in its plural form.

○ Example: On page 1 line 12, there is a singular “liquid biopsy” and then a couple of words down on the same line is the plural form. I believe both should be in the plural form.

• On page 7, the second to last paragraph and last paragraph of the “Enumeration and assessment of CTCs” section glosses over BCa until the very end of line 47. Including an individual sentence before the conclusion paragraph on how there wasn’t any relevant research in BCa found at the current moment might be helpful.

• The “Cell-free DNA and circulation tumor DNA” section has paragraphs that feel very disorganized/detached. It jumps around between the three cancers and the liquid biopsy material. I suggest following the same order of cancers as listed in the other sections, as well as making which liquid biopsy material is being discussed more obvious between sentences.

○Example: On the first paragraph of page 8, most of the paragraph goes over BCa liquid biopsy of urine, but then jumps into PCa liquid biopsy of seminal fluid without indication of the change until the end of the sentence, making it seem disconnected.

• On page 8, line 57, the word “waist” was used, and I believe it was meant to be “waste.”

• On page 9, lines 11-12, the sentence about EVs is awkward grammatically. Instead of “to function in,” remove “to.”

• In the conclusion column of Table 1, the text is currently hard to separate based on the rows of research. Implementing some barriers or separating the text more will help properly divide the information.

---

## [Reviewer Report]

The authors have adequately addressed my critiques. The sections “Enumeration and assessment of CTCs” and “Cell-free DNA and circulation tumor DNA” read well and feel organized when discussing the 3 different urological cancers. All insistences of “Liquid Biopsy” have been changed to its proper form. Table 1 has a nice separation of the rows and is well organized, making it easier to read and understand the material. The manuscript is much improved over the original submission. I have no further critiques for this manuscript and would recommend acceptance by Cambridge Prism: Precision Medicine.